# Removing Inter-Experimental Variability from Functional Data in Systems Neuroscience

**Dominic Gonschorek**[*,1]
dominic.gonschorek@cin.uni-tuebingen.de

**Larissa Höfling**[*,1]
larissa.hoefling@uni-tuebingen.de

**Klaudia P. Szatko**[1]
klaudia.szatko@tuebingen.mpg.de

**Katrin Franke**[1]
katrin.franke@cin.uni-tuebingen.de

**Timm Schubert**[1]
timm.schubert@cin.uni-tuebingen.de

**Benjamin A. Dunn**[2]
benjamin.dunn@ntnu.no

**Philipp Berens**[1]
philipp.berens@uni-tuebingen.de

**David A. Klindt**[‡,2]
klindt.david@gmail.com

**Thomas Euler**[‡,1]
thomas.euler@cin.uni-tuebingen.de

## Abstract

Integrating data from multiple experiments is common practice in systems neuroscience but it requires *inter-experimental variability* to be negligible compared to the biological signal of interest. This requirement is rarely fulfilled; systematic changes between experiments can drastically affect the outcome of complex analysis pipelines. Modern machine learning approaches designed to adapt models across multiple data domains offer flexible ways of removing inter-experimental variability where classical statistical methods often fail. While applications of these methods have been mostly limited to single-cell genomics, in this work, we develop a theoretical framework for domain adaptation in systems neuroscience. We implement this in an adversarial optimization scheme that removes inter-experimental variability while preserving the biological signal. We compare our method to previous approaches on a large-scale dataset of two-photon imaging recordings of retinal bipolar cell responses to visual stimuli. This dataset provides a unique benchmark as it contains biological signal from well-defined cell types that is obscured by large inter-experimental variability. In a supervised setting, we compare the generalization performance of cell type classifiers across experiments, which we validate with anatomical cell type distributions from electron microscopy data. In an unsupervised setting, we remove inter-experimental variability from data which can then be fed into arbitrary downstream analyses. In both settings, we find that our method achieves the best trade-off between removing inter-experimental variability and preserving biological signal. Thus, we offer a flexible approach to remove inter-experimental variability and integrate datasets across experiments in systems neuroscience. Code available at `https://github.com/eulerlab/rave`.

---

[*‡]Equal contributions, [1] University of Tübingen, [2] Norwegian University of Science and Technology.

35th Conference on Neural Information Processing Systems (NeurIPS 2021).

# 1 Introduction

Systems neuroscientists are often concerned with identifying and characterizing how properties of neurons vary along certain dimensions of interest. Differences in these properties between neurons form the basis for sorting them into discrete categories. Both the advance of large-scale data acquisition techniques in experimental neuroscience as well as the development of more efficient and powerful data analysis methods allow collecting and analyzing datasets of increasing size; and hence the discovery of more subtle variations in neural function between cell types [e.g. 1–4]. However, as data acquisition is often an incremental process, it has become common practice to pool data from multiple experiments. This practice ignores variability in the data stemming from external factors, which include non-biological ones (e.g. sample handling resulting in small differences in tissue quality, or temperature fluctuations affecting the rates of biochemical processes) but potentially also unforeseen biological ones (e.g. subtle genetic variations) [3, 5, 6]. Such variability due to external factors, here referred to as *inter-experimental variability*, can confound and obscure the biological signals of interest. In some cases, the source of inter-experimental variability is known and can be modeled [5], but if this is not the case, a method for removing it from the data is required.

The issue of inter-experimental variability in systems neuroscience is analogous to the problem of *domain shift* in machine learning, where the data distribution changes between training ('source') and test ('target') data, causing an algorithm to fail when deployed on data from an unseen target domain [7–10]. Methods that address this issue have to perform some form of *domain adaptation*, i.e. adapting the algorithm to work both on the training as well as some (usually unseen) test domain [11]. In single-cell genomics, a number of different studies have proposed methods for removing inter-experimental variability (see Section 2), but related works in systems neuroscience are lacking, despite the recognized need for such approaches [3, 5]. Here, we contribute to closing this gap as follows:

- We cast the removal of inter-experimental variability from functional data in systems neuroscience in the theoretical framework of domain adaptation (Figure 1 and Section 3).
- We adapt and evaluate different approaches and demonstrate improved performance of cell type assignment, while preserving the biological signals of interest (Table 1 and Figure 4).
- We demonstrate that our method produces cell type predictions on a new dataset that are best aligned with anatomical data (Table 2 and Figure 5).
- Finally, we showcase in a downstream analysis that the corrected data (Figure 3) clearly exhibits biological effects that were obscured by inter-experimental variability (Figure 6).

# 2 Related Work

As mentioned before, few studies have proposed specialized solutions to the issue of inter-experimental variability in systems neuroscience. Two studies have approached the problem of temporal alignment of neural responses across experiments. Zhao et al. [5] proposed a solution to deal with the specific effects of temperature fluctuations on the response kinetics of retinal neurons by modeling them explicitly. Williams et al. [12] proposed a more general method for the temporal alignment of data across trials or recording sessions. Other studies have suggested models of neural function that integrate data across experiments. Shah et al. [3] build encoding models to predict the responses of retinal ganglion cells across different experiments [see also 13] and compare it to covariates such as the gender of an animal. Sorochynsky et al. [14] propose a way to measure noise correlations in each recording and integrate those into models of neural populations of a specific cell type. This latter approach is complementary to our method because it allows the study of the structure and function [see also 15] of noise correlations, which we discard as nuisance variability. Somewhat related to the example application in our paper, Jouty et al. [16] suggested a method to perform non-parametric physiological classification of retinal ganglion cells in the mouse retina while trying to find matching clusters of cell types across experiments. Crucially, all of these approaches offer specialized solutions that do not represent general purpose correction methods.

In single-cell genomics, a number of approaches for removing inter-experimental variability from data have been developed [17–24]. Two such methods are *Harmony* [25] and *scGen* [26]. *Harmony* performs iterative clustering using a variant of soft k-means until convergence to align cells from different datasets in a joint embedding. *scGen*, on the other hand, combines a variational autoencoder

adapted for scRNA-seq data with latent space arithmetics to predict gene expression, while removing inter-experimental variability between datasets. In this paper, we compare our approach against these methods as they have been found to perform particularly well in two benchmarking studies [27, 28].

## 3  Theoretical Framework

The generative process of data denoted by a random variable $X$ with image $\mathcal{X}$ is depicted in Figure 1. The biological signal shared across experiments (e.g. variation due to cell types) is represented by a random variable $S$ ('signal') with image $\mathcal{S}$. We define $D$ ('domain') as a random variable with image $\mathcal{D}$ that represents inter-experimental variability. Now, our objective is to learn a function $f$ that transforms the data into a new random variable $Z := f(X)$ with image $\mathcal{Z}$. Importantly, we distinguish two settings: (*i*) *unsupervised* — where $S$ is unknown and we simply try to retain in $Z$ as much information about the data as possible while removing inter-experimental variability; (*ii*) *supervised* — where $S$ is known and we additionally try to retain in $Z$ as much information about $S$ as possible. These objectives can be formulated in terms of mutual information, giving the unsupervised loss function

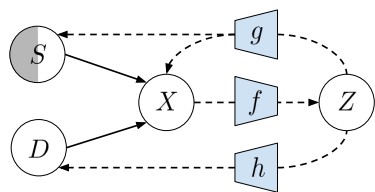

Figure 1: **Problem Setting**. Arrows represent given (solid lines) and modeled (dashed lines) relations. Capital letters denote random variables, small letters transformations (see Section 3). The setting with known $S$ (white circle) is supervised, while unknown $S$ (gray circle) is an unsupervised setting.

$$\mathcal{L} = I(Z; D) - I(Z; X) \tag{1}$$

and, provided knowledge about $S$, we obtain the supervised loss function

$$\mathcal{L}_+ = \mathcal{L} - I(Z; S). \tag{2}$$

Now, $I(Z; D)$ attains its minimum for $I(f(X); D) = 0$ because of the non-negativity of mutual information. And $I(Z; S)$ attains its maximum for $I(f(X); S) = I(X; S)$ because of the data processing inequality. If $f$ were a bijection, it would follow that $I(f(X); S) = I(X; S)$, but also $I(f(X); D) = I(X; D)$. But by assumption, $I(X; D) > 0$ (otherwise there is no inter-experimental variability and we are done) and so we would have $I(f(X); D) > 0$. Thus, at the minimum of $I(Z; D)$, $f$ cannot be a bijection. Generally, if there is an interaction between recording, signal and domain i.e. $I(X; S; D) \neq 0$, then there will be a *trade-off* between maximizing $I(f(X); S)$ and minimizing $I(f(X); D)$. This trade-off becomes even more apparent in the unsupervised setting where $I(f(X); X)$ and $I(f(X); D)$ are clearly competing.

Mutual information quantifies the dependence between two variables but it is difficult to estimate [29–32]. Instead, we measure dependency through nonlinear regression with an appropriate distance metric $d$.[2] Usually $D$ is a discrete random variable indicating the experiment of a recording, and so, to estimate $I(Z; D)$, we can perform classification with a classifier function $h : \mathcal{Z} \to \mathcal{D}$, minimizing the standard cross-entropy $d_{\mathrm{CE}}(h(Z), D)$ (Figure 1). Lemma 10 in [34] shows that this gives a variational lower bound to $I(Z; D)$ [see also 35]. In some cases $S$ may also be discrete (e.g. cell types) and we can do the same, in other cases it might be a (high-dimensional) continuous random variable and so, to approximate $I(Z; S)$, we can perform regression, minimizing the mean squared error $d_{\mathrm{MSE}}(g(Z), S)$. Similarly, in the unsupervised setting, to approximate $I(Z; X)$, we minimize $d_{\mathrm{MSE}}(g(Z), X)$. To keep notation simple, in the unsupervised setting we define the mapping $g : \mathcal{Z} \to \mathcal{X}$, and in the supervised setting $g : \mathcal{Z} \to \mathcal{X} \times \mathcal{S}$. Putting this together, in the unsupervised setting our objective is

$$\min \mathcal{L} \longrightarrow \min_h \max_{g,f} \lambda \, d(h(f(X)), D) - d(g(f(X)), X) \tag{3}$$

where we have introduced a hyperparameter $\lambda$ that mitigates the trade-off discussed above. In the supervised setting our objective becomes

$$\min \mathcal{L}_+ \longrightarrow \min_h \max_{g,f} \lambda \, d(h(f(X)), D) - d(g(f(X)), (X, S)). \tag{4}$$

---

[2]If the joint probability density of two random variables is a bivariate normal distribution, then the mutual information is proportional to their linear correlation [33].

In both equations we find a $\min - \max$ optimization, where $h$ is trying to predict $D$ from $Z$, tightening the lower bound on $I(Z;D)$ [see 34], while $f$ is trying to prevent that by removing information about $D$ from $Z$, effectively lowering $I(Z;D)$. Practically, this optimization scheme has become a standard adversarial setting in machine learning, for instance, in the training of generative adversarial networks [36] or for discriminative adversarial domain adaptation [37].

## 4 Methods

### 4.1 Datasets

To test our model approach, we use two datasets of two-photon imaging recordings [38–40] from the 14 mouse retinal bipolar cell (BC) [41] types' responses to two visual stimuli, a local and full-field chirp stimulus (Figure 3). The axon terminals of BCs stratify at distinct, cell type-specific depths within the second synaptic layer of the retina, the inner plexiform layer (IPL). The functional BC data were obtained by imaging the glutamate output at their axon terminals using the genetically encoded glutamate-sensing fluorescent reporter iGluSnFR [42]. In our study, we refer to these two datasets as *A* [2] and *B* [5] (for further preprocessing see Appendix).

In [2], an anatomy-guided functional clustering approach to group the BCs into the 14 functional types was applied to dataset *A*, thus providing functional reference cell type labels, which do not exist for dataset *B*. However, even if both datasets recorded the same cell types, they suffer from inter-experimental variability making it difficult to match and, for example, to use dataset *A* to predict the cell type labels for dataset *B*. We discuss potential sources of inter-experimental variability in the Appendix. For preprocessing, both BC datasets are high-pass filtered above 0.1 Hz (to remove the trends of decreasing fluorescence signal over time) and resampled to 30 Hz. Each cell's response is normalized to zero mean and standard deviation one. In addition, to ensure high quality responses, only cells with a sufficient response quality are used (for details about quality criterion see [2]).

### 4.2 Models

All methods transform the data, either into a low-dimensional embedding $z \in \mathcal{Z}$ or directly into a reconstruction $\hat{x} \in \mathcal{X}$ from which inter-experimental variability has been removed to a varying degree. Usually we have $\dim(\mathcal{Z}) \ll \dim(\mathcal{X})$ and so, for different downstream evaluations, we map between these representations: (*i*) with least squares reconstructions ($\mathcal{Z} \to \mathcal{X}$), or (*ii*) principle component projections ($\mathcal{X} \to \mathcal{Z}$) (see Appendix).

#### 4.2.1 Unsupervised Model

We parameterize the functions $f, g$ and $h$ (Figure 1) with neural networks. In the unsupervised model, the function $g : \mathcal{Z} \to \mathcal{X}$ provides a reconstruction $\hat{x} := g(z)$, $\hat{x} \in \mathcal{X}$ of the data. With the concurrent task (eq. 3) of minimizing the predictability of the domain $D$, this reconstruction should only contain parts of the original data that are indiscernible across experiments. Since the purpose of our method is to Remove, Adversarially, Variability from datasets collected in different Experiments, we term our model RAVE.

#### 4.2.2 Supervised Model

In the supervised setting, we have partial knowledge about the biological signal $S$. The function $g : \mathcal{Z} \to \mathcal{X} \times \mathcal{S}$ now returns a reconstruction as well as a prediction of that signal $(\hat{x}, \hat{s}) := g(z)$, $\hat{s} \in \mathcal{S}$. When optimizing equation (4), this additional task is equivalent to discriminative adversarial domain adaptation [37]. In the particular data that we work with, we have two datasets $\mathcal{D} := \{A, B\}$, but the biological signal $S$ consists of cell type labels which are only available in the first dataset $A$. Thus, more accurately, this presents a *semi-supervised* scenario where one wishes to classify a newly recorded dataset according to some existing classification scheme. We term this extended version of our model RAVE+.

#### 4.2.3 Training and Optimization Details

All of our models are implemented and optimized in PyTorch [43]. For both RAVE and RAVE+, we use the same model architecture, they only differ in the objective function. We randomly split the data

into training, validation and test set and train all models with empirical risk minimization. Model weights are trained with stochastic gradient descent using one instance of the Adam optimizer [44] for the outer minimization of $f$ and $g$ in equations (3) and (4); and then a second instance of Adam for the inner minimization of $h$ in those equations. We optimize hyperparameters through random search [45] on the validation set and report performances on the test set which is only used for final evaluation. In the random search, we test different learning rates for both optimizers, and also different training schedules. We additionally search over depth, width and drop-out rate for each of the neural networks $(f, g, h)$, as well as the trade-off parameter $\lambda$ introduced in equation (3). Finally, we explore training the inner optimization ($h$, estimating $I(Z; D)$) more often than the outer optimization, which proved more stable and effective in early experiments.

### 4.2.4 Comparison Models

We test three different methods for comparison with our model. Our simplest comparison model (*Linear*) is a linear model that projects out the contrast between the dataset indicator variables (see Appendix). This has an analytic solution and no hyperparameters, serving as a baseline to get an estimate of the correction quality achieved by a standard method in classical statistics. The other two methods (*scGen* and *Harmony*) are run in an unsupervised learning mode without cell type information. Even though *scGen* could be utilized to run in a supervised mode with cell type information, this is not specified in a semi-supervised setting with only partial cell type labels available.

## 4.3 Performance Evaluation

For evaluating the correction performed by the various methods, we analyze their output with respect to dataset-mixing (achieved by removing inter-experimental variability) and preservation of signal information.

### 4.3.1 Dataset-Mixing

The Rand index [46] measures similarity between two clusterings; the adjusted Rand index (ARI) is the Rand index adjusted for chance level (see Appendix) which was recently used by Tran et al. [28] to assess the quality of dataset-mixing in genomics. It takes as input the true and the predicted labels for a set of samples. We define $\text{ARI}_{dom}(z) := \text{ARI}(d, \hat{d}_z)$ with $d$ the original domain labels (of the test set) and $\hat{d}_z$ the domain labels predicted by a classifier trained on $z$. On the raw data ($z = x$), we expect $\text{ARI}_{dom}$ to be high due to inter-experimental variability. After successful correction (with $z$ the output of a model), we expect $\text{ARI}_{dom}$ to be low indicating good dataset-mixing.

In addition, we compute the accuracy ($\text{Acc}_{dom}$) of a domain classifier with the objective to predict the domain labels based on the input data. For low dataset-mixing, we expect a high $\text{Acc}_{dom}$ as it should be trivial for the classifier to differentiate the datasets. However, after removing inter-experimental variability, $\text{Acc}_{dom}$ is supposed to be close to chance level ($\sim 64\%$, cf. Table 1), which would indicate successful dataset-mixing. For the domain classifier, we use a random forest classifier with cross-validated hyperparameters for each model (see Appendix). This is crucial, because a powerful encoder $f$ might hide (through multiple nonlinear transformations) domain information from a simple classifier, but still recover that information in an equally powerful decoder $g$. Conversely, we observe that overly expressive random forest classifiers, tend to overfit on the training set, thus underestimating the preserved domain or type information on the test set.

### 4.3.2 Preservation of Signal Information

In the unsupervised setting, to assess the amount of information preserved about the original data $x$ during the process of removing inter-experimental variability, we evaluate the rank correlation $\text{Corr}(\hat{x}, \hat{x})$ between input $x$ and reconstruction $\hat{x}$. In the (semi-) supervised setting, we have reference cell type labels $s_A$ for dataset $A$. To estimate how much of this information is preserved, we predict cell type labels $\hat{s}_A$ from $z_A$ with random forest classifiers like above (see Figure 2; Appendix for further details).

If a method succeeds at preserving signal information in $z$ after removing inter-experimental variability, then we expect the classifier to have a high accuracy ($\text{Acc}_{type}$). Deteriorating classification

performance between predicting $\hat{s}_A$ from raw data $x_A$ versus predicting it from the model output $z_A$ would indicate signal loss.

Additionally, we would like to evaluate how well cell types can be distinguished and how biologically plausible they are for the unlabeled dataset $B$. To this end, we apply the classifiers to predict cell type labels $\hat{s}_B$ from $z_B$. One direct comparison is between the distributions over cell types in $\hat{s}_B$ and as expected from electron-microscopy (EM) data [47–50] (Figure 5A). However, we can also evaluate the accuracy of these predictions by making use of BC axonal stratification profiles obtained from the same EM data. From those data, we know where in the IPL a BC type stratifies its axon terminals. Thus, we can compare the distribution over IPL depth for the predicted cell types ($\hat{s}_B$) with the distributions expected from EM data. We quantify the difference between the expected and predicted distributions by calculating the Jensen-Shannon distance. We define the depth score (DS) as the mean Jensen-Shannon distance between those two distributions (Table 2). Additionally, we evaluate the robustness of cell type labels $\hat{s}_B$ by fitting the classifier ten times with different seeds and calculating the average ARI between different runs, giving the $\mathrm{ARI}_{type}$ score (Table 1).

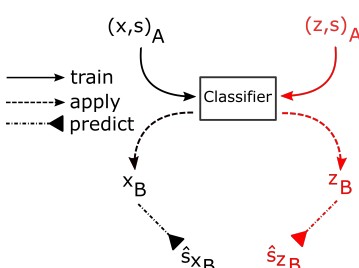

Figure 2: **Workflow.** Evaluating the preservation of signal information: A classifier gets trained on the labeled dataset $A$ (either $(x, s)_A$ or $(z, s)_A$) and applied to dataset $B$ to predict labels (either $\hat{s}_{x_B}$ or $\hat{s}_{z_B}$). The predicted labels are then used for further evaluation.

## 5 Results

### 5.1 Simulation Experiments

First of all, we validated that our model performs as expected on simulated ground truth data. To do this, we generated bipolar cell responses for all 14 cell types based on the published bipolar cell model in Schröder et al. [51]. To simulate different individual neurons, we added small perturbations to the model weights for each cell type until we matched the intra-cell-type variability observed in the real data. Thus, we generated $N = 1000$ distinct neurons for each of the 14 BC types. Approximating the differences of the two datasets in the paper, we presented the model with the slightly altered versions of the stimulus from the actual experiments (see Appendix B). We additionally added white noise to match realistic signal to noise ratios, as estimated from repeated stimulations of the real neurons. This resulted in two datasets ('A' and 'B') with similar intra- and inter-experimental variability as observed in the real data, but with known ground truth cell type labels.

The results are discussed here and, additionally, they are presented in Appendix Fig. 9. We first confirmed that the classifiers are indeed perfectly able to separate these two artificial datasets based on their systematic differences (domain accuracy on raw simulated data: 1.0). However, cell type classifiers trained on dataset A fail completely on dataset B indicating severe inter-experimental variability and a failure to transfer models across datasets (type accuracy on dataset A: 0.98, type accuracy on dataset B: 0.16). In contrast, after correction with RAVE+, the performance of a classifier trained to distinguish the two datasets drops from 1.0 to 0.66 (chance level 0.5), indicating strong removal of inter-experimental variability. Importantly, we find that a classifier trained on the output of RAVE+ on dataset A does now generalize to dataset B and recovers the ground truth cell labels nearly perfectly (type accuracy 0.99). This constitutes an important validation of our model.

### 5.2 Unsupervised Removal of Inter-Experimental Variability

All methods tested (*Linear*, *Harmony*, *scGen*, RAVE and RAVE+) succeed at retaining a significant amount of information about $x$ in $\hat{x}$, reflected by high correlations between $x$ and $\hat{x}$ (Table 1). Corr$(x, \hat{x})$ reaches similar levels for data from both datasets, suggesting that both datasets are modified to find a midway representation. This impression is confirmed when visualizing $x_A$, $x_B$ and $\hat{x}_A$ and $\hat{x}_B$ next to each other (Figure 3). Moreover, we note that a side-effect of the alignment is a more general denoising that RAVE tends to perform along with removing inter-experimental variability. We recognize that this a desirable feature that we will study further in future research.

| Model | $\text{Corr}_A$ ↑ | $\text{Corr}_B$ ↑ | $\text{Acc}_{dom}$ ↓ | $\text{Acc}_{type}$ ↑ | $\text{ARI}_{dom}$ ↓ | $\text{ARI}_{type}$ ↑ |
|---|---|---|---|---|---|---|
| Raw | 100 | 100 | 99.8 (0.1) | 77.4 (0.9) | 99.3 (0.4) | 37.3 (3.3) |
| *Linear* | **99.0** (0.5) | **97.0** (1.7) | 99.5 (0.2) | 83.4 (0.8) | 98.1 (0.7) | 7.6 (1.7) |
| *Harmony* | 72.0 (10.7) | 72.0 (13.8) | 94.2 (0.4) | 82.5 (0.5) | 78.0 (1.6) | 31.4 (2.3) |
| *scGen* | 78.0 (9.8) | 80.0 (10.5) | 99.6 (0.1) | **84.7** (0.8) | 98.7 (0.3) | 14.3 (2.6) |
| RAVE | 60.0 (12.8) | 58.0 (17.3) | 77.5 (0.5) | 69.5 (0.4) | 28.9 (1.2) | 81.2 (2.7) |
| RAVE+ | 59.0 (14.8) | 58.0 (19.1) | **65.9** (0.9) | 78.6 (0.8) | **10.0** (1.2) | **83.7** (2.3) |

Table 1: **Model Comparison.** All entries in percentage. Mean and standard deviation metric scores across 10 random seeds. Bold font in each row indicates best score. $\text{Corr}_A$ ($\text{Corr}_B$) is the correlation of corrected data from dataset $A$ ($B$) with its raw data. $\text{Acc}_{dom}$ ($\text{Acc}_{type}$) is the accuracy of the domain (cell type) classifier. For $\text{ARI}_{dom}$ and $\text{ARI}_{type}$ see Section 4.3.

We show mean traces for exemplary cell types from dataset $A$, and mean traces of cells from dataset $B$ whose cell type labels we predict twice, first based on $x$ (left pathway in Figure 2) and then again based on $\hat{x}$ (right pathway in Figure 2, but on $\hat{x}_{\text{RAVE}}$ instead of $z$). As expected, inter-experimental variability obscuring the common signal $s$ behind $x_A$ and $x_B$ causes the cell type assignment to fail; the similarity between responses of cells assigned to the same cell type, but coming from the different datasets is low (Figure 3, BC type 5t). Repeating the classification pipeline based on $x$ with the same classifier architecture and different seeds yields highly variable cell type predictions for dataset $B$ (Table 1, $\text{ARI}_{type}$) despite high prediction accuracy on dataset $A$ (Table 1, $\text{Acc}_{type}$). This demonstrates a failure in transferring to dataset $B$, and not the classification itself. These results on the raw data $x$ affected by inter-experimental variability were expected; however, the same pattern - low $\text{ARI}_{type}$ (dataset $B$) and high $\text{Acc}_{type}$ (dataset $A$) - is observed for *Harmony*, *scGen* and the *Linear* model. This suggests that these methods fail at removing inter-experimental variability. The high domain accuracy achieved by a classifier trained on the outputs of these models confirms this conclusion. RAVE, on the other hand, succeeds at significantly lowering domain accuracy $\text{Acc}_{dom}$, while at the same time maintaining high scores for $\text{ARI}_{type}$ and $\text{Acc}_{type}$.

## 5.3 Supervised Removal of Inter-Experimental Variability

RAVE+ extends RAVE to the (semi-) supervised setting where (partial) signal information is present. RAVE+ excels at removing inter-experimental variability (Table 1, $\text{Acc}_{dom}$ and $\text{ARI}_{dom}$) and at the same time retaining signal information (Table 1, $\text{Acc}_{type}$ and $\text{ARI}_{type}$). A low dimensional t-SNE

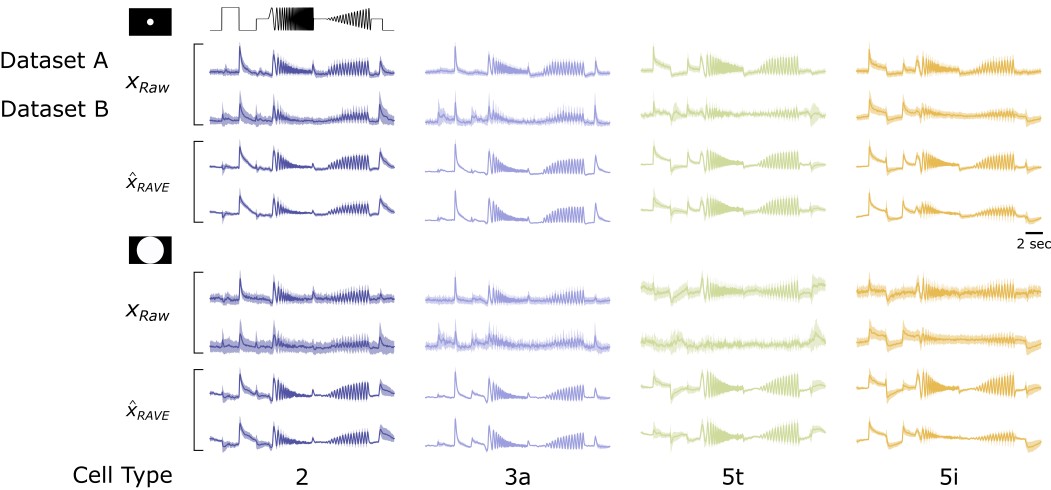

Figure 3: **Exemplary Cell Type Responses from both Datasets to the Chirp Stimuli.** Four bipolar cell type responses of the types 2, 3a, 5t and 5i to the local (top panel) and full-field (bottom panel) chirp of raw data $x_{Raw}$ (two top upper panels) and reconstructed data $\hat{x}_{\text{RAVE}}$ (two bottom lower panels) by RAVE for both datasets $A$ and $B$. Each column shows the mean responses of one cell type (standard deviation shaded).

[52] embedding (Figure 4) shows that cells from datasets *A* and *B* are mapped onto the same cell type "islands". The distribution of types across IPL depth predicted by a classifier trained on $z_{\texttt{RAVE+}}$ matches the expected anatomical distributions better than for all other methods (Figure 5 and Table 2). This provides a valuable validation of the estimate $\hat{s}_B$ learned by RAVE+ in the absence of ground truth knowledge of $s_B$.

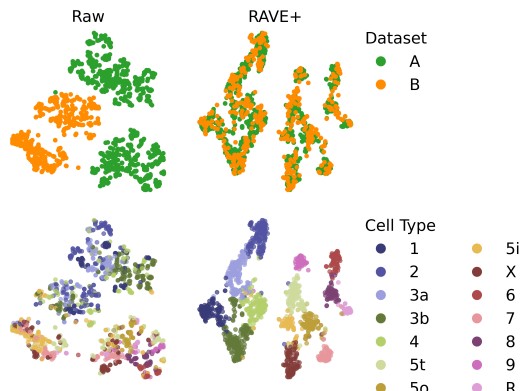

Figure 4: **Dataset Embeddings**. t-SNE embeddings of the test set of raw (left column) and corrected output data by RAVE+ (right column). Embedded cells are color-coded by dataset (top row) and cell type (bottom row). Cell type labels for the raw data of dataset *B* (bottom left) were predicted using a cell type classifier trained on the raw data of dataset *A* (Figure 2, left pathway).

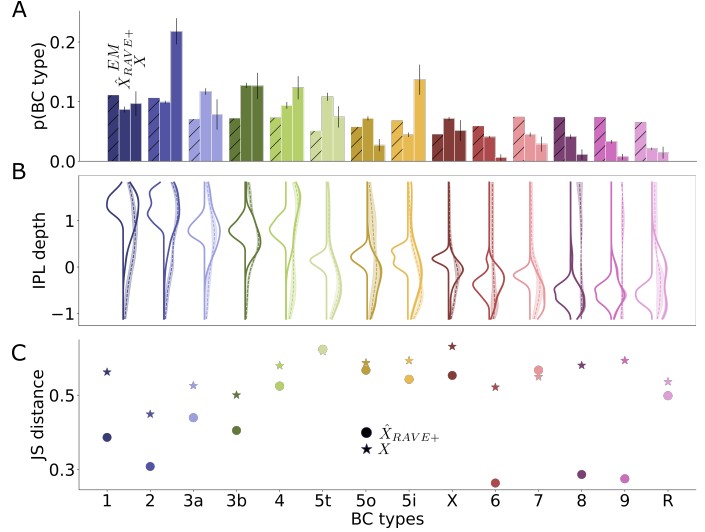

Figure 5: **Distribution Across BC Types and IPL Depths.** We compare the expected and predicted distribution of BCs from dataset *B* across the 14 types and across IPL depth. (**A**) Probability that a BC belongs to a certain type as estimated from EM data; as estimated from BC type labels predicted on $\hat{x}_{\texttt{RAVE+}}$; and as estimated from BC type labels predicted on $x$. Error bars indicate SD across 10 seeds of the classifier. (**B**) Distributions per cell type over IPL depth for EM data (distribution shown to the left), RAVE+ output (solid line to the right) and raw data (dashed line to the right). Shaded area around the distributions shown to the right indicate SD across 10 seeds of the classifier. (**C**) JS distances corresponding to the distributions in B).

## 5.4   Downstream Analyses on Reconstructed Traces

As in our unsupervised setting, it is common that no particular signal information is available and that one wants to remove inter-experimental variability from the data to perform further downstream

| BC Type | 1 | 2 | 3a | 3b | 4 | 5t | 5o | 5i | X | 6 | 7 | 8 | 9 | R | ∅ | all |
|---|---|---|---|---|---|---|---|---|---|---|---|---|---|---|---|---|
| Raw | **34** | 52 | **38** | 42 | 56 | 63 | 62 | 54 | 60 | 52 | **51** | 38 | 58 | **37** | 50 | 31 |
| *Linear* | 58 | 40 | 53 | 51 | 57 | 62 | 57 | 56 | 62 | 55 | 57 | 58 | 58 | 53 | 56 | 34 |
| *Harmony* | 42 | 32 | 42 | 47 | 54 | 59 | 58 | 56 | 61 | 37 | 59 | **26** | 58 | 38 | 48 | 23 |
| *scGen* | 51 | 38 | 48 | 48 | 56 | 59 | **56** | 59 | 63 | 55 | 57 | 56 | 56 | 42 | 53 | 31 |
| RAVE | 41 | 32 | **38** | **40** | 57 | **58** | 62 | 55 | 66 | 55 | 55 | 38 | 50 | 50 | 50 | 23 |
| RAVE+ | 38 | **30** | 43 | **40** | **52** | 62 | **56** | **54** | **55** | **26** | 56 | 28 | **27** | 49 | **44** | **17** |

Table 2: **Depth Score Comparison.** All entries in percentage, lower is better. Bold font in each row indicates best score. Depth Score - Jensen-Shannon (JS) distance between predicted types and EM depth distribution: $JS(p_{EM}(depth|type = t), p_{model}(depth|type = t))$. Last column ("all"): $JS(p_{EM}(type), p_{model}(type))$.

analyses. We show that a previously demonstrated biological effect, obscured by inter-experimental variability in $x$, emerges when performing the same analyses on the reconstructed traces $\hat{x}$ obtained from RAVE. Full-field visual stimulation has been shown to decorrelate responses from different BC types compared to local stimulation due to inhibitory feedback from amacrine cells (see Figure 3A, B in [2]). We expect this fundamental feature to be present in dataset *B*, but cannot fully reproduce it if we assign cells of dataset *B* to cell types based on the raw data (Figure 2, left pathway; and Figure 6A). However, using the reconstructed traces $\hat{x}_{\text{RAVE}}$, the expected feature is unmasked (Figure 2, right pathway, but on $\hat{x}_{\text{RAVE}}$ instead of $z$; and Figure 6B). Here, the mean responses to the local chirp are more correlated across cell types than the full-field responses (Figure 6B, left panel). This can also be seen when comparing the mean correlations between local and full-field chirp responses for each cell type with all other cell types, both of the same and the opposite response polarity (On and Off polarity) (Figure 6B, right panel).

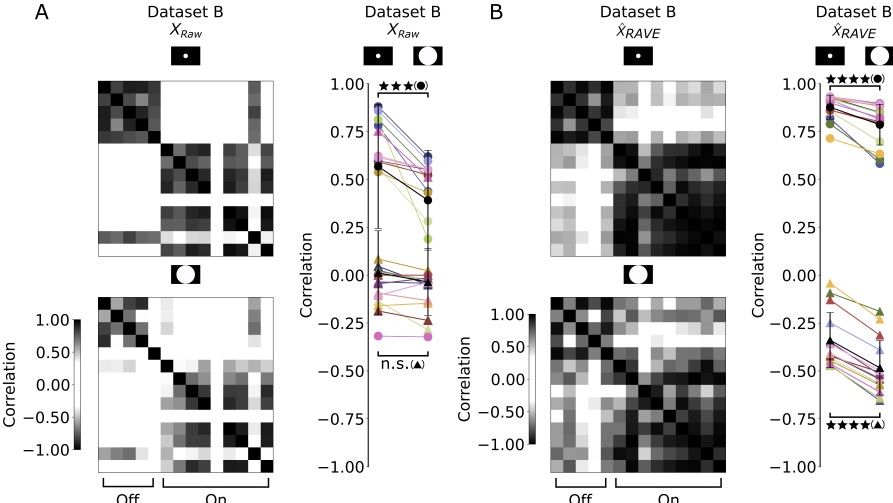

Figure 6: **Removing Inter-Experimental Variability Reveals Biological Feature.** (**A**) Correlation matrices show the correlations between mean responses per cell type to local (top) and full-field (bottom) chirp of raw data $x$ from dataset *B*. The right panel represents the mean correlation for each cell type mean response with all other types of the same (circle) and opposite (triangle) response polarity between local and full-field chirp shown for raw responses. ($x$: mean correlation same polarity: $p_{local} = 0.57$ and $p_{full\text{-}field} = 0.39$, P < 0.005; opposite polarity: $p_{local} = 0.01$ and $p_{full\text{-}field} = -0.04$, n.s.; n = 14, non-parametric paired Wilcoxon signed-rank test).(**B**) Same analysis as **A**, but with the reconstructed responses obtained from RAVE. ($\hat{x}_{\text{RAVE}}$: mean correlation same polarity: $p_{local} = 0.88$ and $p_{full\text{-}field} = 0.79$, P < 0.0005; opposite polarity: $p_{local} = -0.34$ and $p_{full\text{-}field} = -0.48$, P < 0.0005; n = 14, non-parametric paired Wilcoxon signed-rank test).

# 6    Limitations

Our method is limited to datasets where neurons were presented with the same stimulus. For other kinds of data, such as neural recordings from free behavioral paradigms where each trial will be different, it will be difficult to 'align' neural responses in a meaningful way. One solution to this could be to learn a shared embedding space [see 53], from which domain effects are removed, but distinct encoders $f_i$ and decoders $g_i$ for different trials $i$. In another setting, where different stimuli are presented between experiments, one might resort to an approach like Shah et al. [3]. Nevertheless, we do acknowledge that the data in our applications consists of *ex vivo* retinal recordings which have little to no attentional effects or task-dependent noise correlations like they would be present in *in vivo* cortical data. We are optimistic that our framework of adversarially removing inter-experimental variability is still a promising approach in those settings, under the constraint that a much more severe trade-off may need to be made between retaining signal and removing domain shifts.

# 7    Discussion

We present a framework to remove inter-experimental variability from functional recordings in systems neuroscience. To the best of our knowledge, this is the first application of domain adaptation methods to this kind of data. Using our unsupervised (`RAVE`) and (semi-) supervised (`RAVE+`) approaches, we demonstrate that we are able to remove inter-experimental variability while retaining signal information, which allows us to robustly predict cell type labels for a new dataset. We validate those predictions using an anatomy-based comparison to existing EM data.

Furthermore, our unsupervised approach `RAVE` is able to remove inter-experimental variability without cell type information. By using the corrected dataset $B$, we unmask biological effects, obscured by inter-experimental variability, that have been previously described for dataset $A$. Thus, by allowing the integration and alignment of functional recordings across experiments, we show that biological effects in the data become more pronounced when using our model approaches. Inter-experimental variability is ubiquitous and we hope that this method will become a helpful resource to many experimenters as we make the code toolbox publicly available.

We believe that our method can also make a contribution to systems neuroscience research in the context of the 3Rs (Replacement, Reduction and Refinement) for animal ethics: By enabling detection of more subtle biological signals after removal of inter-experimental variability, fewer animals may be needed to test a specific hypothesis. Lastly, we acknowledge that the removal of inter-experimental variability from any kind of data (thus not only within systems neuroscience) can be useful in various applications. Virtually any analysis that aggregates data across experiments can be confounded by inter-experimental variability. Consequently, we cannot exclude the possibility that some military application will find value in this approach. Although unlikely, we cannot fully anticipate such developments. Therefore we condemn, without any exceptions, the use of `RAVE(+)` for any warlike applications or other nefarious purposes.

# 8    Acknowledgments and Disclosure of Funding

This research was funded by the Deutsche Forschungsgemeinschaft (DFG, German Research Foundation) project number 335549539/GRK2381 and the CRC 1233 "Robust Vision" (grant number 276693517). Moreover, this work was partially supported by a Research Council of Norway FRIPRO grant (90532703). PB is a member of the Machine Learning Cluster of Excellence, EXC number 2064/1 – Project number 390727645 and the Tübingen AI Center (FKZ: 01IS18039A). He was supported through a Heisenberg Professorship by the DFG (BE5601/8-1).

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
