# Appendix

## A  Model details

### A.1  Linear mappings between $z$ and $x$

Usually, we have data $x \in \mathbb{R}^{N \times D_1}$ and latent representation $z \in \mathbb{R}^{N \times D_2}$ with $N$ the number of neurons, $D_1$ the dimensionality of the data, $D_2$ the dimensionality of the latent space and, usually, $D_1 \gg D_2$. In cases where a method $m$ does only produce some latent representation $z_m$, we fit a reconstruction $\hat{x}_m = W z_m$ with a least squares projection $W = (z_m^T z_m)^{-1} z_m^T x$. In cases where a method $m$ does only produce some reconstruction $\hat{x}_m$, we produce a simple latent representation $z_m$ by extracting the first $D_2$ columns of the left singular vectors $U$ from the singular value decomposition $x = USV^T$. Both of these projections are fitted on the training data, then fixed and also used on the validation and test data.

## B  Data

We used three datasets, where the first two (dataset *A* [2] n=8417 cells; *B* [54] n=4600) are two-photon recordings of mouse retinal bipolar cell (BC) responses to the chirp stimuli (local and full-field, see [2] for details). Both datasets were used for model fitting and removal of inter-experimental variability. For the validation of cell type predictions made by the different models, we used the third dataset, which comprises EM data of axonal stratification profiles as probability distribution of each BC type [47–50].

The inter-experimental variability between the two functional datasets may originate from, at least, the three following differences between the datasets: (*i*) dataset *A* recorded BCs mostly at certain IPL depths ('ChAT-bands', which are landmarks within the IPL [55]) using tangential scans parallel to the retinal layers, whereas dataset *B* used axial scans employing an electrically tunable lens to record from BCs across the entire IPL simultaneously [5], resulting in different sampling distributions; (*ii*) the chirp stimulus used in dataset *B* differs slightly as the sinusoidal intensity modulation of the increasing frequency is marginally slower; (*iii*) dataset *A* did not employ a gamma correction of the display device to linearize its intensity curve, resulting in slightly different stimulus contrasts [56].

## C  Training Results

The outcome of the random search can be seen in Figure 7, showing metrics on the validation set for both models. To select the best `RAVE` model, we picked the point in the top right corner (center plot, first row, Figure 7). This was the model with the highest $I(Z; X)$, i.e. correlation, and the lowest $I(Z; D)$, i.e. domain classification accuracy. To select the best `RAVE+` model, we picked the (`RAVE+`) point in the top right corner of the $3D$ space spanned by $\{I(Z; X), I(Z; S), -I(Z; D)\}$, i.e. the model with the best reconstruction and cell type prediction accuracy but with the lowest domain prediction accuracy.

Moreover, Figure 7 also demonstrates the trade-off between maximizing $I(Z; X)$ and $I(Z; S)$ and minimizing $I(Z; D)$. In the top row on the left, one can see that models with high $I(Z; X)$ also tend to have a high $I(Z; S)$, indicating that these two tasks can be performed well at the same time (this is what we mean by 'synergy' in the title; naturally, we cannot make a causal statement here). In the top row middle, one can see for models that achieve a high $I(Z; X)$ (some hyperparameter configurations in the random search simple lead to bad models), that there is a negative slope with respect to $I(Z; D)$, indicating that there is a trade-off between optimizing these two objectives. The same can be seen in the top row on the right with respect to $I(Z; S)$ and $I(Z; D)$. The bottom row of Figure 7 zooms in on the high performing models (see axes limits) and indicates the rank correlations. As stated above, we find a positive correlation between $I(Z; X)$ and $I(Z; S)$ (i.e. no conflict), but a negative correlation between $I(Z; X)$ and $I(Z; D)$, and between $I(Z; S)$ and $I(Z; D)$ (i.e. a trade-off).

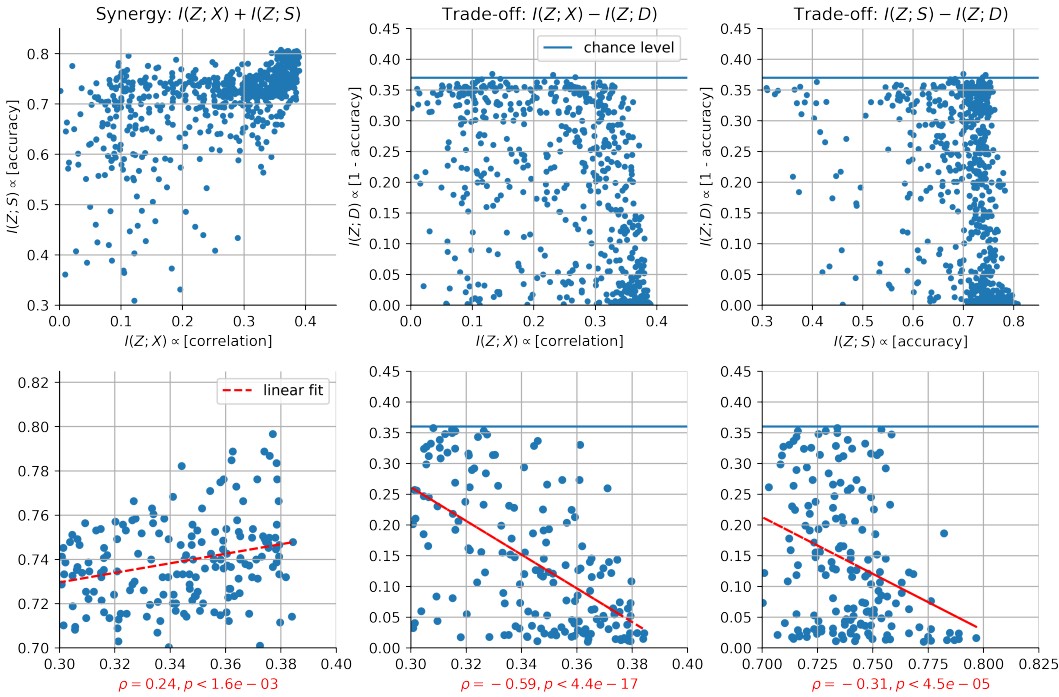

**Figure 7: Random Search Result.** Optimizing models with different hyperparameters shows how the terms in the objective function interact. The top row shows all models, the bottom row only filtered (high-performing) models within the indicated axes ranges. The red lines in the bottom plot indicate linear fits and the red axis labels show the rank correlation coefficients $\rho$ and $p$ values.

# D  Details for Comparison Models

## D.1  Linear Model

Let our full dataset $x \in \mathbb{R}^{(N+M) \times D}$ consists of the concatenated datasets $x_A \in \mathbb{R}^{N \times D}$ and $x_B \in \mathbb{R}^{M \times D}$, i.e. $x = (x_A, x_B)^T$ For the linear model, we chose a design matrix $\beta \in \mathbb{R}^{(N+M) \times 2}$ of the form

$$
\beta = \begin{bmatrix} 1 & -\frac{1}{N} \\ \vdots & \vdots \\ 1 & -\frac{1}{N} \\ 1 & \frac{1}{M} \\ \vdots & \vdots \\ 1 & \frac{1}{M} \end{bmatrix}
\tag{5}
$$

where the first column gives the constant component and the second column (the first $N$ entries equal to $-\frac{1}{N}$ and the second $M$ entries equal to $\frac{1}{M}$) encodes a contrast for the difference between the datasets. The matrix is orthogonal, thus avoiding a singular design. To produce a version of the data with domain effects removed, we fit this to the data with least squares $\gamma = \min_\gamma \|A\gamma - X\|_2^2$, $\gamma \in \mathbb{R}^{2 \times D}$ and project out the second component like

$$
\hat{x}_{Linear} = x - x_{(:,2)}\gamma_{(2,:)}
\tag{6}
$$

to obtain the linearly domain-corrected data.

## D.2  Harmony

For *Harmony*, we used *Harmonypy* (version 0.05) (https://github.com/slowkow/harmonypy), which is the adapted *Harmony* [25] version for the Python environment. As input, we provided a PCA

embedding of the raw data (preprocessed). Here, we used the same number of principle components (PCs) as used for `RAVE`. Since *Harmonypy* returns corrected PCs, we performed further evaluation on these PCs (cf. Appendix Section A.1). To find the best model(s), we performed a random search over hyperparameters. We chose the best model with $\text{Acc}_{dom}$ close to or at chance level, while having high $\text{Acc}_{type}$ on predicted cell type labels. Furthermore, we used the exact same dataset splits as we did for `RAVE` and `RAVE+`.

### D.3 scGen

We used *scGen* [26] (version 2.0.0) within the *Scanpy* [57] (version 1.7.2) working environment. As input to *scGen*, we used the raw responses with dataset source information (either dataset *A* or *B*) using the AnnData [57] object format (version 0.7.6). To run *scGen*, we used the following functions as described in the documentation (https://scgen.readthedocs.io/en/latest/tutorials/scgen_batch_removal.html): $setup\_anndata$ to setup the AnnData object for *scGen*, $SCGEN$ to setup the model, $train$ to train the model and $batch\_removal$ to remove inter-experimental variability.

As *scGen* returns corrected input data, we performed PCA on the output data, which were used for further evaluation (cf. Appendix Section A.1). Here, we used the same number of principle components (PCs) as used for `RAVE`. To find the best model, we performed a random search over hyperparameters. Just like *Harmony*, we chose the best model that had $\text{Acc}_{dom}$ close to or at chance level, while having high $\text{Acc}_{type}$ on predicted cell type labels.

### D.4 Results of Dataset-Mixing by Harmony and scGen

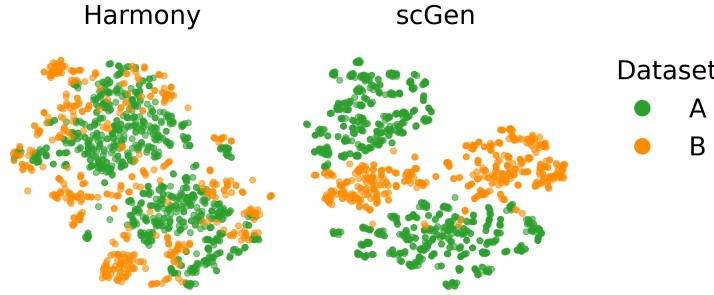

Figure 8: **Dataset Embeddings**. t-SNE embeddings of corrected data by *Harmony* (left) and *scGen* (right). Embedded cells are colored by dataset.

The low dimensional t-SNE embeddings [52, 58] (Figure 8), performed after the application of the two comparison methods (*Harmony* and *scGen*), show that cells from datasets *A* and *B* are not properly mixed; hence they are not removing inter-experimental variability sufficiently (see main paper, Table 1).

## E Simulation experiments

In Figure 9, we present the results of the simulation experiments discussed in the main text. More specifically, we show example simulated cell responses for both stimuli (i.e., datasets 'A' and 'B') in Fig. 9A. Then in Fig. 9B, we demonstrate with a t-SNE embedding that the two datasets show clear inter-experimental variability. However, after correction with `RAVE+`, we can see in Fig. 9C that the two datasets have become aligned, and that the different cell types form clearly separated "islands". And lastly, in Fig. 9D, we see that the depth distributions of the `RAVE+` corrected data are much better aligned with the ground-truth EM distributions than those of the raw data. This last steps further supports our validation procedure for `RAVE+` on real data, based on EM IPL depth profiles.

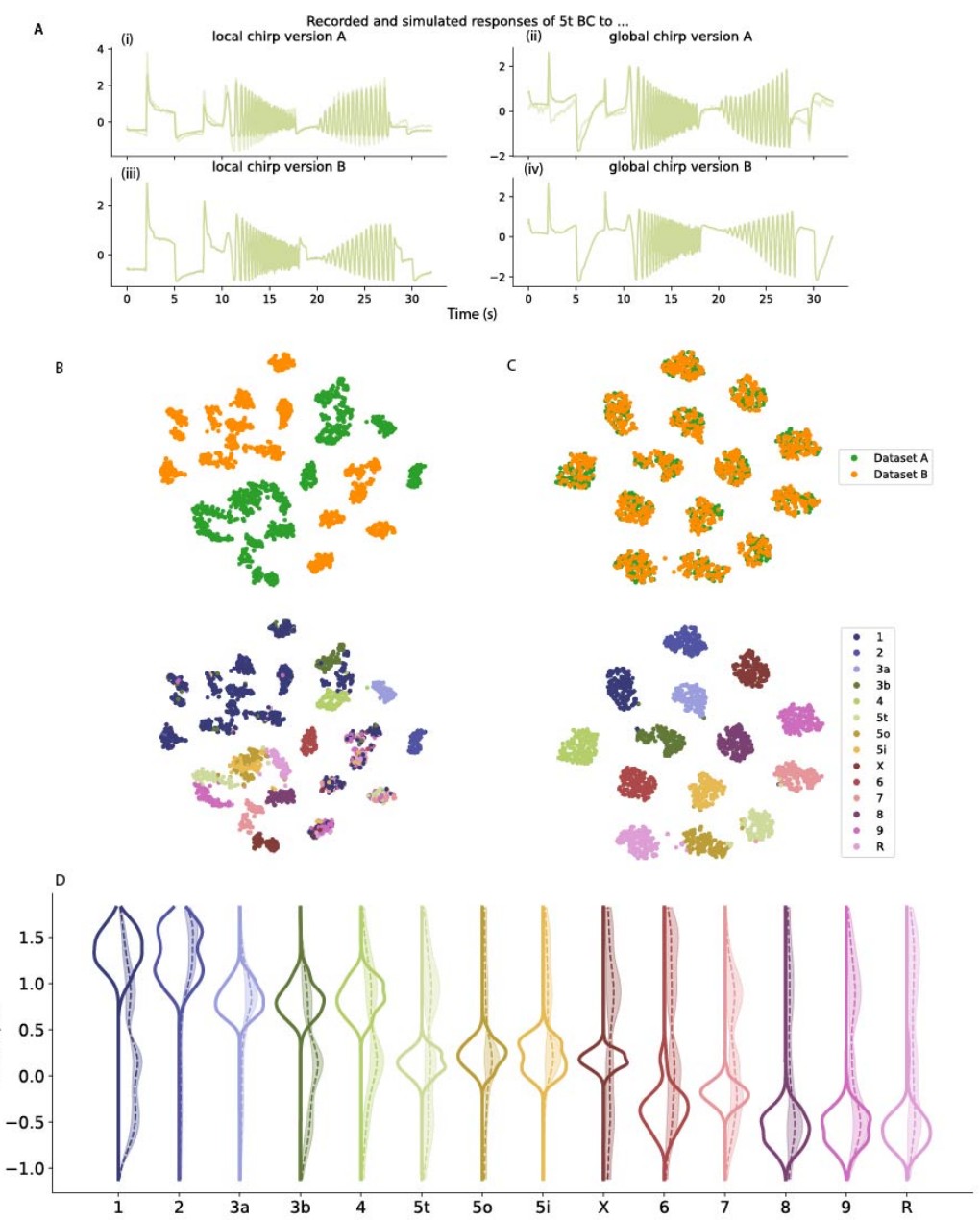

Figure 9: **RAVE+ results on simulated BC responses A**: Simulated (bold) and recorded (light) BC responses of example type 5t in response to (i) the local chirp version A (i.e. the stimulus played in dataset *A*); (ii) the global chirp version A; (iii) the local chirp version B (i.e. the stimulus played in dataset *B*); (iv) the global chirp version B. Note that for chirp version B, we do not have ground truth type labels for recorded responses. **B**: tSNE embedding of raw simulated test set data, colored according to dataset ground truth labels (top) and according to type labels predicted by a classifier trained on raw simulated responses of dataset *A* (bottom). The classifier fails for dataset *B*. **C**: same as **B**, but for RAVE+ output. A classifier trained on RAVE+ output for dataset A achieves accuracies of 1 for dataset *A* and 0.99 for dataset *B*. **D**: Distributions per cell type over IPL depth for EM data (distribution shown to the left), RAVE+ output (solid line to the right) and raw data (dashed line to the right). Shaded area around the distributions shown to the right indicate SD across 10 seeds of the classifier. We sampled IPL depth values for the simulated data according to the type specific distributions known from EM data.

# F  Details for Performance Evaluation

## F.1  Dataset-Mixing

To evaluate dataset-mixing, we used the `scikit-learn` [59] (version 0.24.1) implementation of the adjusted Rand Index (ARI) (cf. [28]).

## F.2  Domain and Cell Type Classifier

In order to evaluate the model correction, we employ a domain and cell type classifier by using a random forest classifier (RFC) [60] from `scikit-learn` with cross-validated hyperparameters for each model. The RFC gets fitted on a subset of dataset *A* and validated on a held-out validation set. We performed the cross-validated grid search on the following hyperparameters: $n\_estimators$ (5, 10, 20, 30), $max\_depth$ (5, 10, 15, 20, `None`), $ccp\_alpha$ (0, 0.001, 0.01) and $max\_samples$ (0.5, 0.7, 0.9, 1). The grid search was performed using 10 random seeds to avoid overfitting (see main paper, section 4.3.1) and the best scoring RFC (highest $Acc_{type}$; lowest $Acc_{dom}$ on validation set, respectively) was selected to predict cell types or domain labels on the test set of the corrected data.

## F.3  Visualization of Dataset Embedding

We used the t-SNE algorithm [52] to visualize the cells in a low dimensional space [58]. For this purpose, we chose the `openTSNE` [61] implementation (version 0.6.0) in Python and ran it with default parameters and fixed seed.