# OpenReview forum: "Removing Inter-Experimental Variability from Functional Data in Systems Neuroscience"
_NeurIPS.cc/2021/Conference — NeurIPS 2021 Spotlight_

### Official Review · Reviewer_YJrn · 2021-07-13

**Rating:** 8
**Confidence:** 4

**Summary:**

This paper introduces a new framework to remove non-biological variability to facilitate data aggregation across studies. The problem is viewed as a transformation of the input data, such that the transformed output is not informative of the “domain” (e.g., experimental condition or study) while simultaneously capturing relevant information from the data. This high-level procedure is implemented via an adversarial training strategy, where the mutual information terms of the objective function are approximated via a standard cross-entropy or MMSE loss. The authors validate their method using two murine datasets, in which the neural recording procedures are slightly different; the proposed framework outperforms the standard Harmony and scGen in the supervised and unsupervised settings.

**Limitations And Societal Impact:**

The authors discuss two key limitations of their approach: (1) the experimental paradigm must be the same across datasets, and (2) the method does not work well if the data is noisy (e.g., whole-brain fMRI). These limitations do hinder the societal impact of this particular work, particularly in human neuroimaging studies. With that said, the need for data aggregation methods is key in many application domains, so the authors are working on a critical problem with broad impact.

**Main Review:**

Overall, this is a solid paper that presents an interesting new twist on data fusion. The authors complement their mathematical framework with a straightforward implementation using GANs. The evaluation is also comprehensive and clearly demonstrates the promise of the method. With that said, there are a few questionable assumptions. Specific comments on the paper are as follows:

Strengths

+ The paper tackles the ubiquitous yet challenging problem of removing non-biological data variability.
+ The authors propose a simple and elegant framework for nuisance removal that relies on distributional properties of the data.
+ The authors develop both a supervised and unsupervised version of the model, thus allowing it to be used in a variety of application domains.
+ The experimental results are comprehensive. The authors compare their method (RAVE and RAVE+) with state-of-the-art approaches, and in each case, numerous performance metrics are extracted.

Weaknesses

- In the case where S (signal of interest) is continuous, the authors approximate the mutual information between Z (transformation) and S via a MSE objective. Depending on the dimensionality and characteristics of S, the MSE optimum can be quite far from the MI optimum.
- The training setup, model architecture, data processing, and hyperparameter tuning should be described in the main text and not buried in the Appendix. The authors have almost 1/2 page of free space.
- The closing sentence of the paper (condemning the use of this method in warlike applications) is puzzling. How would RAVE or RAVE+ be used for such purposes? More details are needed.

**Time Spent Reviewing:**

3 hours

---

> ### Author Response · Authors · 2021-08-10
> **Initial response to review**
>
> We really appreciate your positive assessment of our work and agree that data aggregation across experiments is an important problem in systems neuroscience and beyond. We provide a point-by-point response to your questions and suggestions below. As instructed in the NeurIPS guidelines for authors, we will upload the changed manuscript after the review process.
>
> Thank you for the suggestion to move the information about training setup, model architecture, data processing and hyperparameter tuning from the Appendix to the main text of the manuscript. We have moved implementation details required for reproducibility to the methods section and taken care that it does not affect the clarity and reading flow of the manuscript. Overall, we agree that this information may be crucial for the reader and should already be included in the main text.
>
> You are correct, approximating mutual information (MI) with nonlinear regression and a mean-squared-error (MSE) objective will in some cases be misaligned with the MI optimum. We have considered more advanced approaches for estimating mutual information (e.g. Alemi et al., [2017](https://arxiv.org/pdf/1612.00410.pdf)). However, we empirically observed that the parsimonious MSE objective, corresponding to a normality assumption on the residuals of our nonlinear regression, yielded strong results.
>
> Regarding the closing sentences [“Lastly, we acknowledge the fact that any machine learning method or approach can be utilized for malevolent purposes. Therefore, we condemn, without any exceptions, any use of RAVE(+), in any manner whatsoever, for the usage of weapons or warlike applications.“], the main aspect that we wanted to address with this statement is that the removal of inter-experimental variability from any kind of data (thus not only within systems neuroscience) can be useful in various applications. Virtually any analysis that aggregates data across experiments can be confounded by inter-experimental variability. Consequently, we cannot exclude the possibility that some military application will find value in this approach. Although unlikely, we cannot fully anticipate such developments. As a minimal protection against such nefarious purposes, we decided to add that statement to the paper to condemn and dissociate ourselves from any such applications. We have reformulated to make this motivation more clear in the manuscript.
>
> We realise that our limitations section might have been slightly misleading. We mentioned the detrimental effects of noise correlations that our approach would blindly try to remove, while they might constitute a functional aspect of the system’s response (see also response to reviewer 5unW). In contrast, data with increased observational noise, such as human fMRI recordings, would actually present a good application case for our model. The reason for this is that beyond removing inter-experimental variability our model also performs more general denoising (see e.g. the reconstructions in Fig. 3). We recognize that this is a desirable feature and we will make sure to mention it in the paper.

---

> > ### Comment · Reviewer_YJrn · 2021-08-29
> > **Rebuttal**
> >
> > Thank you for the thoughtful response to my comments.

---

### Official Review · Reviewer_Ra6g · 2021-07-14

**Rating:** 7
**Confidence:** 4

**Summary:**

In the paper "Removing Inter-Experimental Variability from Functional Data in Systems Neuroscience", the authors propose a novel framework for alleviating inter-experimental variability. They present a supervised as well as an unsupervised variant of their approach and evaluated the framework on cell type datasets. They further compared their results to EM data, giving further credibility to their findings.

**Limitations And Societal Impact:**

The authors consider limitations of their work in a dedicated section where they consider the problem of different stimuli. They consider benefits with respect to the 3Rs which I find convincing.

**Main Review:**

Contrary to what the authors said in their checklist, they did not include code or data in their submission! I believe not all of the datasets are publicly available. In the manuscript, the authors promise to make the code available upon acceptance.

Originality:
The idea to combine datasets by dealing with inter-experimental variability is not new. The authors do a reasonable job of summarizing related work. For systems neuroscience, they refer to Zhao et al. as well as Shah et al. However, their discussion of related work is not complete. In Sorochynskyi, Deny, Marre and Ferrari, PLoS Comput Biol 2021, the authors proposed a method involving statistical models of noise correlations to deal with inter-experimental variability. I think this is relevant and should have been discussed.
The authors have applied methods from machine learning for domain adaptation. For all I know such methods have not been applied to systems neuroscience before.

Quality:
The authors used state-of-the-art methods from domain adaptation. They introduce their framework based on concepts from information theory. This is done in a rigorous way. The demonstration of their framework is done on real data where they could validate their results by means of EM data. This is convincing and impressive.

Clarity:
The clarity of the manuscript is outstanding. The text is very well written, motivates the problem well and introduces the framework in a formal yet easily accessible way. The figures are also very good and easy to follow.

Significance:
From the systems neuroscience perspective, the proposed methods are of great importance. Inter-experimental variability is a profound problem due to natural data limitations occurring in neuroscience. A systematic and flexible way to remove this variability opens up new possibilities for data analysis.
I am not an expert in domain adaptation, so I cannot assess whether this is a contribution that is valuable to the machine learning for domain adaptation community as well.

**Time Spent Reviewing:**

5

---

> ### Author Response · Authors · 2021-08-10
> **Initial response to review**
>
> Thank you for your extensive review and positive feedback. We are happy to see that you consider our manuscript “rigorous” and “convincing” and that you find it of “great importance” for system neuroscience. Below, we tried to include your suggestion to further improve the quality and potential impact of our work. As instructed in the conference email, we will upload the changed manuscript after the review process.
>
> You are completely right that we did not include the code and/or data in our submission and we would like to apologise for this oversight. We have rectified this situation and are happy to share a [link](https://we.tl/t-Zp31oLzN3C)* where we have now properly anonymized our code and included a demonstration notebook that showcases the functionality of our model and the analysis to obtain our additional results. We will release the actual datasets used in the paper upon publication in compliance with institutional proprietary regulations. We genuinely care about reproducibility and, more specifically, turning this into an easy-to-use tool for system neuroscientists. Thus, we are planning to release a documented github repository with demos and step-by-step tutorials. We hope that this further improves the usefulness of our work for the community.
>
> We agree with the concern that the related work section was not complete with regard to the systems neuroscience literature. We will extend our manuscript with citations and discussion of the following papers (see response to reviewer j9Ur): Jouty et al. ([2018](https://www.frontiersin.org/articles/10.3389/fncel.2018.00481/full)), Williams et al. ([2020](https://www.sciencedirect.com/science/article/pii/S0896627319308943)) and Sorochynskyi et al. ([2021](https://journals.plos.org/ploscompbiol/article?id=10.1371/journal.pcbi.1008501)). Specifically, we thank you for the reference to Sorochynskyi et al. ([2021](https://journals.plos.org/ploscompbiol/article?id=10.1371/journal.pcbi.1008501)) which we have missed in our literature search. We will discuss this in our related work and believe that it offers an interesting alternative to our research direction. While in our work we consider any inter-experimental variability as a nuisance that should be discarded, studying the precise structure but also functional role (see also Ecker et al. [2011](https://www.nature.com/articles/npre.2011.6170.1.pdf?origin=ppub)) of noise correlations provides an important complementary approach for the understanding of neural information processing systems.
>
> \* The information about IPL depth distribution per cell type originates from the following two publications:
> Kim, J. S., Greene, M. J., Zlateski, A., Lee, K., Richardson, M., Turaga, S. C., … Seung, H. S. (2014). Space-time wiring specificity supports direction selectivity in the retina. Nature, 509(7500), 331–336. https://doi.org/10.1038/nature13240
> Greene, M. J., Kim, J. S., & Seung, H. S. (2016). Analogous Convergence of Sustained and Transient Inputs in Parallel On and Off Pathways for Retinal Motion Computation. Cell Reports, 14(8), 1892–1900. https://doi.org/10.1016/j.celrep.2016.02.001

---

> > ### Comment · Reviewer_Ra6g · 2021-08-29
> > **Thank you for your response**
> >
> > It is good to hear that the authors will release the datasets and the code upon publication and that they will revise their related work section. Overall, I believe that my original rating is still appropriate for this work.

---

### Official Review · Reviewer_5unW · 2021-07-16

**Rating:** 7
**Confidence:** 3

**Summary:**

The authors develop a method for removing inter-experimental variability in systems neuroscience experiments. The general idea is to learn a new representation that maximizes the information retained about the data (and in the supervised case, also the signal of interest), while minimizing the information about the individual experiments. To do this, they use an adversarial training approach, where one network tries to remove information about the individual experiments while retaining total information in the new representation, and another network tries to predict the individual experiment from this new representation. They demonstrate their approach on experimental 2-photon recordings of retinal bipolar cells, recorded in 2 different experiments.

**Limitations And Societal Impact:**

Yes

**Main Review:**

Quality:

Overall, I found the quality of the paper high. It is technically sound, and compared to a few alternative methods on an experimental neuroscience dataset.

It would have been nice (perhaps in the appendix), to show that the method can successfully recover the ground truth when it is known in Simulations.


Originality:
As the authors state, they are using an existing general approach within machine learning of adversarial training to deal with domain shifts. That being said, I agree it is novel in its application to systems neuroscience - at least I am not aware of anyone who has used this approach.


Clarity:

I found the clarity to be mixed in this paper.

In particular, I found it challenging to follow the description of the model. For instance:
-Why does Eq 3 include an S - isn’t this unsupervised? Should the S say X?  And how do you get from Eq 1 to Eq 3 - is the order of the terms flipped?
-I would be more explicit that minimizing d is maximizing information (if I am properly understanding). This relates to my above point that it would be helpful to more clearly walk through going from Eqs 1 and 2 to 3 and 4.

I also found myself frequently looking back and forth between different parts of the paper, which made it more difficult to read/understand.
-It would be helpful to have a greater description of the figures in the Figure legends, or have the figures closer to where they’re being described in the text
-When reading the Methods, I didn’t know how it was going to be used or why it mattered, and then in the Results, there was no quick description of any of the methods being used
(so I needed to go back and forth). Some synthesis would be helpful.


Significance:

I think the authors are addressing an important problem of controlling for inter-experiment variability, which is rarely addressed. I think this general approach could be very useful in systems neuroscience.

While I feel this could be useful in theory, I wasn’t very convinced of the usefulness towards answering scientific questions based on the demonstrations in the paper. For instance, what would be the reason for a researcher to do the analysis in Fig. 6 based on a dataset with unknown cell types (besides demonstrating that your method works) - wouldn’t you just try to address this question in datasets where you actually have cell labels?


Other:

Line 141 - what is phi? (and why is it necessary to introduce new  nomenclature here?)

Fig. 4 - in the bottom panel, how are you coloring dots for dataset B, since you don’t know those cell types?

**Time Spent Reviewing:**

7

---

> ### Author Response · Authors · 2021-08-10
> **Initial response to review**
>
> Thank you for your thoughtful review and valuable feedback. We appreciate that you found our paper “technically sound” and of “high quality”. Below, we tried to include your suggestions to improve the paper further and make it a useful resource for this important problem in systems neuroscience. As instructed in the conference email, we will upload the changed manuscript after the review process.
>
> We strongly agree with you that additional experiments based on simulated ground truth data constitute an important confirmation of the functionality of our model. Therefore, we generated bipolar cell responses for all 14 cell types based on the published bipolar cell model in Schröder et al. ([2020](https://proceedings.neurips.cc/paper/2020/hash/b139e104214a08ae3f2ebcce149cdf6e-Abstract.html)). To simulate different individual neurons, we added small perturbations to the model weights for each cell type until we matched the intra-cell-type variability observed in the real data. Thus, we generated N=1000 distinct neurons for each of the 14 BC types. Approximating the differences of the two datasets in the paper, we presented the model with the slightly altered versions of the stimulus from the actual experiments (see Appendix). This resulted in two datasets (‘A’ and ‘B’) with similar intra- and inter-experimental variability as observed in the real data, but with known ground truth cell type labels.
>
> The results are presented [here](https://ibb.co/Gd59bJ8). We first confirmed that the classifiers are indeed perfectly able to separate these two artificial datasets based on their systematics differences (domain accuracy on raw simulated data: 1.0). However, cell type classifiers trained on dataset A fail completely on dataset B indicating severe inter-experimental variability and a failure to transfer models across datasets (type accuracy on dataset A: 0.98, type accuracy on dataset B: 0.16). In contrast, after correction with RAVE+, the performance of a classifier trained to distinguish the two datasets drops from 1.0 to 0.66 (chance level 0.5), indicating strong removal of inter-experimental variability. Importantly, we find that a classifier trained on the output of RAVE+ on dataset A does now generalize to the dataset B and recovers the ground truth cell labels nearly perfectly (type accuracy 0.99). This constitutes an important validation of our approach and we will add this section to the paper.
>
> Thank you for your suggestions to further improve the clarity of our paper. We agree with you that there is always room for improvement. We are committed to the idea that our work should be accessible and useful for a broad set of readers from different backgrounds. More specifically, we implemented the following clarifications and changes:
> 1. We swapped the order of the terms in equations 3 and 4 to make the correspondence with equations 1 and 2 more clear.
> 2. You are correct, the ‘S’ in equation 3 should be an ‘X’. We have changed this. (Moreover the domains of g in lines 109 & 110 should, of course, be Z not X).
> 3. We now denote the cell type labels ‘s’ (instead of ‘c’ as before in lines 212 ff.) to be consistent with the theory section.
> 4. Line 141 phi: We agree that there is no need for new nomenclature here, we reformulated without introducing this additional function. (In a similar spirit we removed the superfluous psi in lines 213 ff.).
> 5. We also updated figure legends to be more self-contained, while remaining concise.
> 6. We synthesized the methods and results section in the sense that they become more self-contained (with quick reminder descriptions) to reduce going back and forth and facilitate the readability of the paper.
> 7. We moved the figures closer to their mentioning in the text.
>
> Fig. 4: To identify cell types for the raw dataset B, we make use of the cell type classifier (Fig. 2) by training it on the raw data of dataset A and making predictions for dataset B. Thus, this also illustrates the failure of transferring information from dataset A to dataset B due to the presence of inter-experimental variability since they are not properly clustered and segregated. An alternative option would have been to exclude the cell types for dataset B, but we decided against this for the sake of completeness.
>
> Thank you for your question about the demonstration in Fig. 6. We agree with you that our approach is potentially useful for any analysis where naive aggregation of data across experiments obscures effects. While the demonstration of the usefulness of our method in Fig. 6 may seem slightly contrived, we basically wanted to show a non-trivial example where our method works. Furthermore, this analysis allowed us to test the output of RAVE in a setting where the expected outcome is known, such that it could serve as a validation of RAVE’s ability to preserve and reveal underlying biological effects.
>
> Again, thank you very much for reviewing our paper and for your valuable suggestions!

---

> > ### Comment · Reviewer_5unW · 2021-08-26
> > **Thank you for your response**
> >
> > Thank you for your thorough response, the additional experiments, and the clarifications. I am increasing my score to reflect these additions.

---

### Official Review · Reviewer_j9Ur · 2021-07-21

**Rating:** 8
**Confidence:** 3

**Summary:**

The paper casts the removal of inter-experimental variability from functional data in systems neuroscience in the theoretical framework of domain adaptation. The paper adapts various approaches and subsequently demonstrate improved performance of cell type assignment. The paper also shows that the proposed theoretical framework can produce predictions that are best aligned to the anatomical data, as well as effectively reduces inter-experimental variability to reveal unobscured biological effects.

**Limitations And Societal Impact:**

Yes

**Main Review:**

The paper clearly delineates related work in the area and provides various citations for the reader to refer to. The discussion is acceptable in terms of detail, but not excellent. To be an excellent , it would be useful to cite some more related works and (optionally) make a small comparison table to delineate similarities/differences of each and how the current proposed method addresses these limitations. The methodology is presented clearly with easy-to-understand equations and diagrams. In addition, the method is compared to three other baselines (two of which were mentioned in the recent works). The performance between the different methods make it clear that the current methodology is superior in accuracy, IPL depth, and identifying JS divergence, compared to the other methods, showing success in reducing inter-experiment variability. The paper also explicitly shows a nice example that demonstrates that their method can be used to reveal biological effects previously unnoticed when the dataset is contaminated with inter-experiment variability. Limitations and a clear discussion of the results/ethical implications are provided. Overall, the paper is solid work, and proves to be a good candidate for acceptance to Neurips.

**Time Spent Reviewing:**

0.5 hrs

---

> ### Author Response · Authors · 2021-08-10
> **Initial response to review**
>
> Thank you for your positive review and feedback. We appreciate that you find our work “solid” and “presented clearly”. To further improve the quality of the paper, we included your suggestions below. As instructed in the conference email, we will upload the changed manuscript after the review process.
>
> We appreciate your recommendations for improving the discussion. We just wanted to clarify whether you are proposing to cite more related work in the discussion? We think that this would fit more naturally in the related work section. However, we do not insist on a specific formatting order and are open to your suggestions.
>
> As suggested, we extended our manuscript by citing and discussing the following papers: Jouty et al. ([2018](https://www.frontiersin.org/articles/10.3389/fncel.2018.00481/full)) who try to perform non-parametric physiological classification of retinal ganglion cells in the mouse retina while trying to find matching clusters of cell types across experiments; Williams et al. ([2020](https://www.sciencedirect.com/science/article/pii/S0896627319308943)) who demonstrate non-parametric temporal alignment of neural responses across multiple experiments; and Sorochynskyi et al. ([2021](https://journals.plos.org/ploscompbiol/article?id=10.1371/journal.pcbi.1008501)) who built models of cell type specific noise correlations from neural responses recorded sequentially across multiple experiments. The last reference offers an interesting complementary approach as we discuss in our response to reviewer Ra6g.
>
> Additionally, we also followed your suggestion to compile a table with all methods. One difficulty in doing so is that the existing approaches are designed for rather different applications: temporal alignment (Zhao et al., [2020](https://www.nature.com/articles/s41598-020-60214-z); Williams et al., [2020](https://www.sciencedirect.com/science/article/pii/S0896627319308943)), predicting neural responses (Shah et al., [2020](https://www.biorxiv.org/content/biorxiv/early/2021/02/14/2021.02.14.431169.full.pdf); Sorochynskyi et al. [2021](https://journals.plos.org/ploscompbiol/article?id=10.1371/journal.pcbi.1008501)), clustering neurons (Jouty et al. [2018](https://www.frontiersin.org/articles/10.3389/fncel.2018.00481/full)), genomics (Lotfollahi et al. [2019](https://idp.nature.com/authorize/casa?redirect_uri=https://www.nature.com/articles/s41592-019-0494-8&casa_token=CNowlz_4lOMAAAAA:cjl0hN1xQYLtVPKj9y5BXJ8fz6CfOVzosLWpg46_Urdffbzivvst5jnW_M9nUAUyDBsyt7C2wUibJQew); Korsunsky et al., [2019](https://idp.nature.com/authorize/casa?redirect_uri=https://www.nature.com/articles/s41592-019-0619-0&casa_token=nISg0j59AWsAAAAA:8XJOxwnTEVloRfEfgKbzNymjLeQde_aODbzhXJBkR1RT2ihwSeqNZ38AhhKz_GzoY0_6Tb31u19y-3Ud)) and general removal of inter-experimental variability (ours). Consequently, there are very few shared features that could be compared by quantification or by noting their presence/absence. This resulted in a table with mostly textual descriptions. Therefore, we think that the discussion of these different approaches is better formatted and more easily digestible in an expanded related work section.

---

> > ### Comment · Reviewer_j9Ur · 2021-09-10
> > **reviewer reply**
> >
> > No problem! It was my pleasure to read your work. Your additions look great. It is fine to keep in the related works section. That would be the more natural fit.

---

### Decision · Program_Chairs · 2021-09-27

**Decision:**

Accept (Spotlight)

**Comment:**

This paper provides a method for removing inter-experimental variability from functional datasets arising in systems neuroscience. All reviewers agreed that the results are significant and the paper was well written and executed.